# The Relation between Wavefunction and 3D Space Implies Many Worlds with Local Beables and Probabilities

Ovidiu Cristinel Stoica

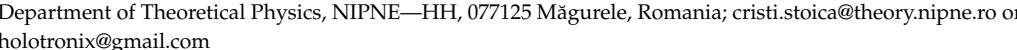

Department of Theoretical Physics, NIPNE—HH, 077125 Măgurele, Romania; cristi.stoica@theory.nipne.ro or holotronix@gmail.com

**Abstract:** We show that the quantum wavefunctional can be seen as a set of classical fields on the 3D space aggregated by a measure. We obtain a complete description of the wavefunctional in terms of classical local beables. With this correspondence, classical explanations of the macro level and of probabilities transfer almost directly to the quantum. A key difference is that, in quantum theory, the classical states coexist in parallel, so the probabilities come from self-location uncertainty. We show that these states are distributed according to the Born rule. The coexistence of classical states implies that there are many worlds, even if we assume the collapse postulate. This leads automatically to a new version of the many-worlds interpretation in which the major objections are addressed naturally. We show that background-free quantum gravity provides additional support for this proposal and suggests why branching happens toward the future.

**Keywords:** wavefunction; 3D space; many-worlds interpretation; Born rule; branch counting; wavefunctional formulation of quantum field theory; quantum gravity; background-independence

## 1. Introduction

This article explores the relationship between the wavefunction and 3D space in quantum mechanics. This relation should be clarified because even in nonrelativistic quantum mechanics (NRQM), the wavefunction is not a function defined on 3D space but on the higher dimensional configuration space. Apparently, the situation does not seem to improve in more sophisticated theories, such as quantum field theory (QFT) or quantum gravity (QG). We will see that the answer to this question touches several foundational questions in quantum mechanics and suggests that a version of the many-worlds interpretation gives the answers.

It is important to understand the wavefunction in terms of fundamental entities having a clear 3D space ontology, i.e., entities that are *in* or *on* 3D space. J.S. Bell calls such entities local beables [1]. We will work with quantum fields in the wavefunctional formulation of quantum field theory. Because the configuration space consists of fields instead of positions, the wavefunction is replaced by a wavefunctional. In Sections 2 and 3, we will see how the wavefunctional has a natural interpretation as many classical fields on 3D space aggregated by a measure. This answers the following

**Question 1.** *Can the wavefunction encode local beables or be described in terms of them?*

We use this in Section 3 to propose answers to the following related question:

**Question 2.** *What is the ontology of the wavefunction?*

The answer is "a set of classical fields aggregated by a measure". The phases become absorbed in the U(1) gauges of the classical fields, so this also addresses the question:

**Question 3.** *Why is the wavefunction a complex function?*

The classical fields determine a basis of the Hilbert space. Because they fully consist of local beables, we call the states from the resulting basis ontic states. The ontic states are compatible with the macrostates in which the universe is observed to be. Then, building on Sections 2 and 3, Section 4 deals with the questions:

**Question 4.** *How do* 3D *objects in space arise from the wavefunction?*

**Question 5.** *Why does the world look classical at the macroscopic level?*

Most wavefunctionals describe macroscopic superpositions. Therefore, to answer Question 5, it is important to understand which of the wavefunctionals do not describe such superpositions. In classical physics, this problem does not exist precisely because all classical entities are local beables. This indicates that local beables should give the answer in quantum theory too. We argue that it does: microstates have to belong to a basis (determined by the classical fields) whose states will be called ontic.

In classical physics, since ultimately the results of experiments are examined at the macro level, the fact that a macrostate corresponds to more possible microstates is the key to explain how probabilities arise. This is a problem in quantum theory:

**Question 6.** *How do probabilities arise in quantum theory?*

The key difference is that in quantum theory, the sample space seems to depend on the experiment. In Section 5, we will see that, in quantum theory, the relation between wavefunctional and 3D space leads to a unique sample space for all experiments if we understand that, ultimately, all observations are macroscopic. Therefore, the answer is similar to the classical one, but thinking that subsystems are separate systems obfuscates this because the ontic basis only exists for the total system, not for subsystems.

We will see that, while in classical physics, probabilities describe the agent's ignorance of the actual microstate of the system, in quantum theory, they represent the ignorance of the agent's self-location in one of many microstates. This leads to a derivation of the Born rule and the meaning of probabilities by "counting" the ontic states per macrostate.

In Section 6, it is shown that if we assume wavefunction collapse, probabilities encounter severe difficulties. Whether we assume wavefunction collapse or not, multiple ontic states have to exist simultaneously. This suggests as the natural interpretation a version of the many-worlds interpretation (MWI) [2–4] that results from this analysis. This addresses

**Question 7.** *How should we interpret quantum mechanics?*

This version of MWI includes probabilities in the classical sense due to the distribution of microstates per macrostate rather than by simply interpreting the squared norm of the state vector as a probability, as it is often proposed. In Section 7, our derivation of the Born rule is compared with other possible ways to count microstates or worlds.

In Section 8, we will see that strong additional support for these findings comes from background-free quantum gravity (which includes most approaches to QG). In the background-free approaches, most linear combinations of states with different 3D geometry cannot represent superpositions. This leads to the dissociation of the state into states with different classical geometries, practically forcing upon us a new version of MWI.

When applied to the Big Bang, this dissociation effect suggests an answer to the time asymmetry of the branching structure problem of the MWI (Section 9):

**Question 8.** *Why does branching happen toward the future and not also toward the past?*

These results address the major objections against the MWI, in a very conservative and classical-like manner. The big picture resulting from this analysis will be discussed in Section 10.

Several technical details were relegated in Appendices A–C to simplify the article.

## 2. The Wavefunctional and the 3D Space

Let $\Sigma$ be the 3D space, which is usually a manifold. If we ignore the curvature due to gravity, we can assume that $\Sigma = \mathbb{R}^3$, but this works for any 3D manifold and even for discrete structures.

Intuitively, we expect that an object is in 3D space if it can be seen as consisting of parts, each of them having a definite position in 3D space. For example, a function or a field defined on a space can be recovered from its values at different positions.

Strictly speaking, a point or set of points from 3D space is *in* 3D space. A classical field $\phi$ is *on* 3D space, in the sense that $\phi$ is a function on the 3D space $\Sigma$, $\phi : \Sigma \to \mathcal{S}$, where $\mathcal{S}$ is a set in which the field is valued. For example, $\mathcal{S}$ can be $\mathbb{R}$ or $\mathbb{C}$ for real or complex scalar fields, $\mathbb{R}^3$ for real vectors, etc. More generally, a field is a section in a fiber bundle over $\Sigma$. For example, the field $\phi : \Sigma \to \mathcal{S}$ is a section of the trivial bundle $\Sigma \times \mathcal{S} \stackrel{\pi_1}{\mapsto} \Sigma$, where $\pi_1$ is the projection on $\Sigma$. The field $\phi$ is a section in the sense that $\pi_1 \circ \phi = 1_\Sigma$, where $1_\Sigma$ is the identity map of $\Sigma$.

In Appendix A, it is explained that the wavefunction is, in fact, an object of 3D space geometry, and that it can even be faithfully represented as infinitely many fields on $\Sigma$ that have a local Hamiltonian evolution. However, the representation that will be used in this article comes directly and naturally from quantum field theory (QFT).

In the Schrödinger wavefunctional formulation of QFT, the configuration space $\mathcal{C}$ consists of classical fields [5]. Therefore, instead of a wavefunction, one uses a function of functions or fields, a wavefunctional $\Psi[\phi]$, $\Psi : \mathcal{C} \to \mathbb{C}$.

There are more types of classical fields to be quantized, which can be scalar, spinor, vector, or tensor fields. They can also have internal degrees of freedom, corresponding to the internal spaces of gauge symmetries. Let $\phi = (\phi_1, \ldots, \phi_n)$ contain all the components of all these fields. The operators $\widehat{\phi}_j(x)$ act by multiplication with $\phi_j(x)$. Their canonical conjugates are the functional derivatives $\widehat{\pi}_j(y) := -i\hbar\delta/\delta\phi_j(y)$. They satisfy the canonical commutation relations if they are bosonic and the canonical anticommutation relations if they are fermionic, in which case they are Grassmann numbers. We assume that the manifold $\mathcal{C}$ is endowed with a measure $\mu$ (see Appendix B for a discussion of its existence).

The Hilbert space $\mathcal{H}$ consists of the $\mu$-measurable functionals $\Psi : \mathcal{C} \to \mathbb{C}$ that are square-integrable with respect to the measure $\mu$,

$$\mathcal{H} := L^2(\mathcal{C}, \mu, \mathbb{C}). \tag{1}$$

The state vectors labeled by $\phi \in \mathcal{C}$ form an orthogonal basis $(|\phi\rangle)_{\phi \in \mathcal{C}}$ so that, for any compact-supported continuous functional $\Psi : \mathcal{C} \to \mathbb{C}$,

$$\int_\mathcal{C} \langle\phi|\phi'\rangle \Psi[\phi'] \mathcal{D}\mu(\phi') = \Psi[\phi]. \tag{2}$$

The time evolution of the universe is governed by the Schrödinger equation:

**Postulate 1** (Unitary evolution). *The state of the universe can be represented by a unit vector* $|\Psi(t)\rangle \in \mathcal{H}$, *whose evolution is described by the equation*

$$|\Psi(t)\rangle = \widehat{U}_{t,t_0}|\Psi(t_0)\rangle. \tag{3}$$

Here, the unitary evolution operator $\widehat{U}_{t,t_0} := e^{-\frac{i}{\hbar}(t-t_0)\widehat{H}}$ between the times $t_0$ and $t$ is determined by the time-independent selfadjoint operator $\widehat{H}$, called the Hamiltonian.

The Hamiltonian operator acts locally in 3D space [5]. The wavefunctional formulation allows the recovery of the usual formulation of QFT in terms of operator-valued distributions and of the Fock representation [5].

The wavefunctional $\Psi$ can be understood naturally as consisting of a number $|\mathcal{C}|$ (usually infinite) of fields on 3D space of the form $(\phi, c_\phi)$, where $\phi \in \mathcal{C}$, $\phi : \Sigma \to \mathbb{C}$ is a classical field from $\mathcal{C}$, $c_\phi := \Psi[\phi]$ is constant in space, and $|\mathcal{C}|$ is the cardinal of $\mathcal{C}$. Then, $\Psi$ is equivalent to a classical field $\Psi : \Sigma \to \mathbb{C}^{2|\mathcal{C}|}$ on the 3D space $\Sigma$,

$$\Psi(x) = \left(\phi(x), c_\phi\right)_{\phi \in \mathcal{C}}. \tag{4}$$

This representation follows directly from the wavefunctional formulation. In the next section, we will see how the phase of $c_\phi$ can be absorbed in $\phi$ and that this allows us to interpret the microstate of the universe as a classical field with a given gauge. We will see that $\Psi$ can be understood as a densitized set of gauge classical fields. For this reason, we call the basis $(\phi)_{\phi \in \mathcal{C}}$ the ontic basis.

## 3. The Wavefunction'S Ontology: A Densitized Set of Classical Worlds

Let us write down the wavefunctional $\Psi$ in polar form with $r[\phi] \geq 0$,

$$\Psi = \int_\mathcal{C} r[\phi] e^{i\theta[\phi]} |\phi\rangle \mathcal{D}\mu[\phi]. \tag{5}$$

We assume that there is a global $U(1)$ gauge symmetry so that at least one of the classical fields $\phi_j$, $j \in \{1, \ldots, n\}$ transforms nontrivially under global $U(1)$ gauge transformations. We know that this is true in our universe because there are always electromagnetic potentials and Dirac fields. Although $U(1)$ acts differently on different types of fields, for simplicity, we denote by $\phi \mapsto e^{i\theta}\phi$ the gauge transformation of the classical field $\phi$. Then, $\phi \neq e^{i\theta}\phi$ for any $\theta$ that is not an integer multiple of $2\pi$.

A global gauge transformation of a classical field $\phi$ results in a physically equivalent field $e^{i\theta}\phi$. On the other hand, a multiplication of the vector $|\phi\rangle$ with $e^{i\theta}$ results in a physically equivalent vector $e^{i\theta}|\phi\rangle$. Then, without loss of consistency, we can identify

$$e^{i\theta[\phi]}|\phi\rangle = |e^{i\theta[\phi]}\phi\rangle. \tag{6}$$

In other words, a phase change in $|\phi\rangle$ is made equivalent to a $U(1)$ gauge transformation of $\phi$. This is physically consistent because the physical state remains unchanged under these transformations. The commutative diagram (7) summarizes this.

$$
\begin{array}{ccc}
\phi_\gamma & \xrightarrow{\text{gauge transformation}} & e^{i\theta}\phi_\gamma \\
\downarrow{\scriptstyle\text{quantization}} & & \downarrow{\scriptstyle\text{quantization}} \\
|\phi_\gamma\rangle & \xrightarrow{\text{phase transformation}} & e^{i\theta}|\phi_\gamma\rangle = |e^{i\theta}\phi_\gamma\rangle
\end{array}
\tag{7}
$$

Since $U(1) \cong SO(2, \mathbb{R})$, the complex numbers $e^{i\theta[\phi]}$ from Equation (5) can be interpreted as real gauge transformations, answering Question 3.

Since the configuration space $\mathcal{C}$ was constructed by fixing a gauge, a gauge transformation leads to a different configuration space $\widetilde{\mathcal{C}}$ and a different ontic basis $(\widetilde{\phi})_{\phi \in \widetilde{\mathcal{C}}}$,

$$\widetilde{\phi} := e^{i\theta[\phi]}\phi \text{ and } \widetilde{\mathcal{C}} := \{\widetilde{\phi}|\phi \in \mathcal{C}\}. \tag{8}$$

However, Equation (6) shows that the resulting Hilbert space is independent of the gauge coefficients $\theta[\phi]$ from Equation (8) that define the configuration space $\widetilde{\mathcal{C}}$, $\mathcal{H} = L^2(\widetilde{\mathcal{C}}, \mu, \mathbb{C})$.

On the other hand, from Equations (6) and (8), $\Psi$ in the form from Equation (5) becomes a real functional in the basis $(\widetilde{\phi})_{\phi \in \widetilde{\mathcal{C}}}$ because the phases are absorbed,

$$\Psi = \int_{\widetilde{\mathcal{C}}} r[\widetilde{\phi}] |\widetilde{\phi}\rangle \mathcal{D}\mu[\widetilde{\phi}]. \tag{9}$$

Since $r[\phi]$ has to be a $\mu$-measurable function on $\mathcal{C}$, there is a measure $\widetilde{\mu}$ so that

$$\mathcal{D}\widetilde{\mu} = r[\phi]\mathcal{D}\mu[\phi]. \tag{10}$$

Then, from Equations (6) and (10), Equation (5) becomes

$$\Psi = \int_{\widetilde{\mathcal{C}}} |\widetilde{\phi}\rangle \mathcal{D}\widetilde{\mu}[\widetilde{\phi}]. \tag{11}$$

Equation (8) explains complex numbers in quantum theory, addressing Question 3. Representation (11) addresses Question 2, by suggesting the following ontology of the wavefunctional: it consists of ontic states combined according to a density.

## 4. The World Appears Classical at the Macroscopic Level

At the macroscopic level, the observers have imperfect "resolution" so that states that are microscopically different cannot be distinguished. We assume that this defines an equivalence relation of states. Classical macrostates are equivalence classes of classical states from the configuration space $\mathcal{C}$, and they form a (disjoint) partition of $\mathcal{C}$,

$$\mathcal{C} = \bigsqcup_{\alpha \in \mathcal{A}} \mathcal{C}_\alpha. \tag{12}$$

This induces a direct sum decomposition of the Hilbert space $\mathcal{H}$ defined in Equation (1),

$$\mathcal{H} = \bigoplus_{\alpha \in \mathcal{A}} \mathcal{H}_\alpha, \mathcal{H}_\alpha := L^2(\mathcal{C}_\alpha, \mu, \mathbb{C}). \tag{13}$$

**Definition 1.** *In the following, the subspace $\mathcal{H}_\alpha$ will represent macrostates. The states represented by vectors from macrostates will be called quasiclassical states. Projectors $\widehat{\mathsf{P}}_\alpha$ on subspaces representing macrostates, so that $\mathcal{H}_\alpha = \widehat{\mathsf{P}}_\alpha \mathcal{H}$, will be called macroprojectors.*

**Postulate 2** (Macroclassicality). *(i) If the state of the universe is $|\Psi\rangle$, the world is observed to be in a macrostate $\widehat{\mathsf{P}}_\alpha \mathcal{H}$ for which $\widehat{\mathsf{P}}_\alpha|\Psi\rangle \neq 0$. (ii) Subsequent observations are consistent with the state of the universe being $\widehat{\mathsf{P}}_\alpha|\Psi\rangle / |\widehat{\mathsf{P}}_\alpha|\Psi\rangle|$ at that time.*

If Postulate 2 seems too complicated, it is because it carefully avoids assuming more than can be observed. In particular, it avoids presuming whether the wavefunction collapses or not. For quantum measurements, it avoids assuming too much about the state of the "observed subsystem", because what we actually observe is a macrostate in which the pointer observable has a definite state.

It is useful to detail how Postulate 2 applies to quantum measurements. Let $\mathcal{H}_S$ be the Hilbert space of the observed system. Let $\widehat{\mathsf{A}}$ be a Hermitian operator on $\mathcal{H}_S$, representing the observable of interest, with eigenbasis $(\psi_1^{\mathsf{A}}, \ldots, \psi_n^{\mathsf{A}})$. To indicate the result of a measurement, the measuring device contains a pointer, which is readable at the macroscopic level and can be found in one of the eigenstates $(\zeta_0^{\mathsf{A}}, \zeta_1^{\mathsf{A}}, \ldots, \zeta_n^{\mathsf{A}})$ of the pointer observable $\widehat{\mathsf{Z}}^{\mathsf{A}}$. Let $\zeta_0^{\mathsf{A}}$ represent the "ready" state of the pointer, and $|\psi\rangle$ the state of the observed system before the measurement. If the measurement of $\widehat{\mathsf{A}}$ takes place between $t_0$ and $t_1$, Equation (3) leads to a linear combination involving pointer states,

$$|\Psi(t_1)\rangle = \widehat{U}_{t_1,t_0}|\psi\rangle \otimes |\zeta_0^{\mathsf{A}}\rangle \otimes \ldots = \sum_j \langle \psi_j^{\mathsf{A}}|\psi\rangle |\psi_j^{\mathsf{A}}\rangle \otimes |\zeta_j^{\mathsf{A}}\rangle \otimes \ldots \tag{14}$$

Since the pointer eigenstates are macroscopically distinguishable, the states $|\psi_j^A\rangle \otimes |\zeta_j^A\rangle \otimes \ldots$ are quasiclassical and correspond to distinct macrostates. Therefore, a state containing the pointer in an eigenstate of $\widehat{Z}^A$ is quasiclassical, as stated in Postulate 2.

Postulate 2 accommodates the possibility of different measurement setups that we would normally consider incompatible. The measuring devices that perform different measurements are macroscopically distinct, so the macrostates corresponding to different measurement results are orthogonal. The incompatibility is between the observables associated with the observed subsystem, but the possible macrostates from which we normally infer the state of the observed subsystem are orthogonal.

The possible resulting states of the universe are not determined by the eigenstates of the observed system nor by those of the pointer of the measuring device. A pointer is a macroscopic object, and it corresponds to the macrostates of the universe, but each macrostate consists of a continuum of microstates from $(|\phi\rangle)_{\phi \in \mathcal{C}}$. The world should be in a definite ontic state. This suggests the following

**Postulate 3** (Microstates). *Only the ontic states $(|\phi\rangle)_{\phi \in \mathcal{C}}$ can be microstates.*

At first sight, there is a tension between Postulate 2, which says that the future observations are consistent with the state being $\widehat{P}_\alpha|\Psi\rangle / |\widehat{P}_\alpha|\Psi\rangle|$, and Postulate 3, which says that microstates can only be from $(|\phi\rangle)_{\phi \in \mathcal{C}}$. However, what Postulate 3 says is that each macrostate consists of microstates that are ontic states, $\widehat{P}_\alpha = \int_{\mathcal{C}_\alpha} |\phi\rangle\langle\phi| \mathcal{D}\mu[\phi]$. This is consistent with Postulate 2, since $\widehat{P}_\alpha|\Psi\rangle = \int_{\mathcal{C}_\alpha} \Psi[\phi] \mathcal{D}\mu[\phi]$.

Postulate 3 is consistent with Postulate 2, because the classical states $|\phi\rangle$ are also quasiclassical, since each $\phi$ belongs to a unique macrostate $\widetilde{\mathcal{C}}_\alpha$. It also clarifies Postulate 2: the world looks classical because its microstates are classical ontic states. Since the ontic states consist of objects in 3D space, this addresses Questions 4 and 5.

In standard quantum mechanics (SQM), the Projection Postulate was introduced to explain why we observe only one of the states $|\psi_j^A\rangle \otimes |\zeta_j^A\rangle \otimes \ldots$. The Projection Postulate was given in terms of quantum measurements [6,7]. Here, we replaced the Projection Postulate with Postulate 2, which

- is more general, including measurements as particular cases,
- avoids presuming whether the wavefunction collapses or not,
- relates the macrostates to microstates of the form $|\phi\rangle$, where $\phi \in \mathcal{C}$ have clear relations with 3D space.

The probabilities are given by the Born rule:

**Rule 1** (Born rule). *If the state of the universe is represented by $|\Psi\rangle$, the probability that an observation of the world finds it in the macrostate $\widehat{P}_\alpha \mathcal{H}$ is*

$$P_\alpha = \langle\Psi|\widehat{P}_\alpha|\Psi\rangle. \tag{15}$$

From Equation (11) $\widehat{P}_\alpha|\Psi\rangle = \int_{\widetilde{\mathcal{C}}_\alpha} |\widetilde{\phi}\rangle \mathcal{D}\widetilde{\mu}[\widetilde{\phi}]$, therefore,

$$\left|\int_{\widetilde{\mathcal{C}}_\alpha} |\widetilde{\phi}\rangle \mathcal{D}\widetilde{\mu}[\widetilde{\phi}]\right|^2 = \langle\Psi|\widehat{P}_\alpha|\Psi\rangle. \tag{16}$$

This is not yet a proof of the Born rule. In SQM, the Born rule is postulated, but in Section 5, we will derive it based on the relation between Postulates 2 and 3.

## 5. Naive Counting Gives the Born Rule in the Continuous Limit

Suppose Alice asks Bob to participate in the following experiment. Alice instructs Bob to wait until a bell rings and as soon as the bell rings, to push a button. The button stops a stopwatch, and Bob, without reading it, has to guess whether the stopwatch indicates an even or an odd number for the millisecond.

A way to interpret the probability that Bob assigns to the event is that the state of the universe contains the state of the stopwatch, including its property that the millisecond is an even or an odd number. Bob does not know the state of the world, but he can attribute the probability 1/2 to the event that the millisecond is even. This subjective probability is based on the incomplete knowledge of the state of the system.

Another interpretation is that Bob is a succession of infinitely many instances, one for each moment of time. There is an instance of Bob which stops the stopwatch as a result of (a previous instance of Bob) hearing the bell ringing. Then, (a subsequent instance of) Bob can interpret the probability as representing the odds that his instance that pressed the button was located along the time axis in an interval labeled by an even or an odd number representing the millisecond. This is the self-location probability of Bob in time.

In the example with the stopwatch, both the subjective view and the self-location view are valid. However, an adept of presentism may prefer the subjective view, while an adept of eternalism may prefer the self-location view of probability.

Now consider an experiment in which Alice sends Bob a qubit in the state $1/\sqrt{2}$ $(|0\rangle + |1\rangle)$, asking him to determine whether the qubit's state is $|0\rangle$ or $|1\rangle$. The probability that Bob determines that the qubit is in the state $|1\rangle$ is 1/2. However, the subjective view applies if the wavefunction collapses, while if both worlds exist, the probability comes from Bob's ignorance of whether he is the Bob instance in the world where the result is $|0\rangle$ or the one in which the result is $|1\rangle$, so the self-location view applies.

Now, let $(|\phi_k\rangle)_{k\in\{1,...,n\}}$ be orthonormal eigenvectors of the operator $\widehat{A}$ representing the observable, and $\mathcal{H}_S$ the observed system's Hilbert space. Or $(|\phi_k\rangle)_{k\in\{1,...,n\}}$ can be an orthogonal system of quasiclassical states, and $\widehat{A}$ a macroscopic observable. Then, if

$$|\psi\rangle = \frac{1}{\sqrt{n}} \sum_{k=1}^{n} |\phi_k\rangle \tag{17}$$

is the state vector of the observed system, and $\widehat{P}_j$ is the projector of the eigenspace corresponding to the eigenvalue $\lambda_j$, the Born rule coincides with counting states:

$$\langle\psi|\widehat{P}_j|\psi\rangle = \frac{1}{n} \sum_{|\phi_k\rangle \in \widehat{P}_j \mathcal{H}_S} \langle\phi_k|\phi_k\rangle = \frac{n_j}{n}, \tag{18}$$

where $n_j$ is the number of the eigenbasis vectors $|\phi_k\rangle$ that are eigenvectors for $\lambda_j$.

However, this "naive state counting" does not give the right probabilities because it coincides with the Born rule only in this special situation. In general, the coefficients in Equation (17) are distinct complex numbers, and counting them will give a different probability from the Born rule. For this reason, in the standard versions of MWI it was proposed to interpret self-location uncertainty as being given by the squared amplitude and not simply by counting [8], and even that this should be postulated [9].

However, the worlds are not determined by the vectors $|\phi_k\rangle$. What is naive about the "naive self-location view" is to count the eigenstates of the observed system or of the pointer state as worlds in which the observer can be located. The full ontic states should be counted, and an agent should be in a definite ontic state. Self-location should be about the possible ontic states of the universe, which are $(|\phi\rangle)_{\phi\in\mathcal{C}}$ (Postulate 3).

Moreover, while counting states works only for states of the form (17), in the continuous limit, it works for all states $|\Psi\rangle \in \mathcal{H}$. However, counting should be applied to the whole system, not to its parts (Postulate 2), and only to ontic states (Postulate 3).

**Theorem 1.** *The Born rule is obtained as the continuous limit of counting ontic states.*

**Proof.** The macroprojectors consistent with Postulate 3 have the form

$$\widehat{P}_\alpha = \int_{\mathcal{C}_\alpha} |\phi\rangle\langle\phi|\mathcal{D}\mu[\phi], \tag{19}$$

where the set $\mathcal{C}_\alpha \subset \mathcal{C}$ is $\mu$-measurable. For any unit vector $|\Psi\rangle \in \mathcal{C}$, there is an infinite sequence $(\mathcal{P}_n)_{n\in\mathbb{N}}$ of sets of projectors with the following properties:

(i) Each projector from $\mathcal{P}_n$ has the form $\widehat{\mathsf{P}}_{n,k} = \int_{D_{n,k}} |\phi\rangle\langle\phi|\mathcal{D}\mu[\phi]$, where $\mathcal{C} = \bigsqcup_{k=1}^{2^n} D_{n,k}$ is a partition of $\mathcal{C}$ into measurable subsets so that $\int_{D_{n,k}} r^2[\phi]\mathcal{D}\mu[\phi] = 1/2^n$.

(ii) For each $n$, $\mathcal{P}_{n+1}$ refines $\mathcal{P}_n$, i.e., projectors from $\mathcal{P}_n$ are sums of those from $\mathcal{P}_{n+1}$.

(iii) The measure of the sets $D_{n,k}$ included in $\mathcal{C}_\alpha$ converges to the measure of $\mathcal{C}_\alpha$.

Then, from (i) and (ii), for each $n$, $|\Psi\rangle$ decomposes as $|\Psi\rangle = 1/\sqrt{2^n} \sum_{k=1}^{2^n} |n,k\rangle$, where $|n,k\rangle := \sqrt{2^n}\widehat{\mathsf{P}}_{n,k}|\Psi\rangle$ are orthogonal unit vectors. From (iii), the sequence $(\mathcal{P}_n)_{n\in\mathbb{N}}$ converges to a refinement of the set of macro-projectors $(\widehat{\mathsf{P}}_\alpha)_{\alpha\in\mathcal{A}}$. Hence, the continuous limit of a counting as in (18) gives the Born rule. For more details, see [10]. □

Then, due to Postulate 3, the Born rule is obtained as a probability measure over the ontic states. This is possible because $\widetilde{\mathcal{C}}$ becomes a sample space, $(\widetilde{\mathcal{C}}_\alpha)_{\alpha\in\mathcal{A}}$ an event space, and $P : (\widetilde{\mathcal{C}}_\alpha)_{\alpha\in\mathcal{A}} \to [0,1]$, $P(\widetilde{\mathcal{C}}_\alpha) = \int_{\widetilde{\mathcal{C}}_\alpha} r^2[\widetilde{\phi}]\mathcal{D}\mu$ a probability function. Therefore, $\left(\widetilde{\mathcal{C}}, (\widetilde{\mathcal{C}}_\alpha)_{\alpha\in\mathcal{A}}, \widetilde{\mathcal{C}}_\alpha \mapsto \int_{\widetilde{\mathcal{C}}_\alpha} r^2[\widetilde{\phi}]\mathcal{D}\mu\right)$ becomes a classical probability space. At any instant in time, the probability density $|\Psi[\phi]|^2$ on $\widetilde{\mathcal{C}}$ can be interpreted similarly to the probability density on the phase space from classical physics. If only one microstate exists, but it is unknown, the probability is subjective. If more microstates can coexist simultaneously, it can be interpreted as self-location probability. This answers Question 6.

## 6. Wavefunction Collapse Is Inconsistent with Our Derivation of the Born Rule

It may seem that we can interpret Equation (16) probabilistically in two different ways and get the Born rule (15). The subjective view applies if there is only one world whose microstate is unknown to the agent, and the wavefunction collapses to be consistent with Postulate 2. The self-location uncertainty view applies if there are many worlds, but the agent does not know in which of them they are located.

Now we will see that SQM, which assumes wavefunction collapse, is inconsistent with Postulate 3 and, therefore, with our derivation of the Born rule. In SQM, $|\Psi(t)\rangle$ is a microstate at all times. Whenever it evolves into a linear combination over more macrostates it collapses to one of them to ensure consistency with Postulate 2.

However, if there is only one world that collapses to avoid macroscopic superpositions, it should be allowed to be in states that do not belong to the same basis. To see this, let us look again at Equation (14). It assumes that at $t_0$

$$|\Psi(t_0)\rangle = |\psi\rangle \otimes |\zeta_0^{\mathsf{A}}\rangle \otimes \ldots . \tag{20}$$

The vector $|\psi\rangle \in \mathcal{H}_S$ can be any unit vector from $\mathcal{H}_S$. Let $|\psi'\rangle \in \mathcal{H}_S$ be another unit vector. Then, the total state vector is $|\Psi'(t_0)\rangle = |\psi'\rangle \otimes |\zeta_0^{\mathsf{A}}\rangle \otimes \ldots$. In particular, there are vectors $|\psi\rangle, |\psi'\rangle \in \mathcal{H}_S$ so that, at $t_0$, $\langle\psi|\psi'\rangle \neq 0$, which implies $\langle\Psi(t_0)|\Psi(t_0)'\rangle \neq 0$. The states $|\Psi(t_0)\rangle$ and $|\Psi'(t_0)\rangle$ are distinct microstates of the same macrostate in which the pointer state is $|\zeta_0^{\mathsf{A}}\rangle$. Since, in SQM, the world is allowed to be in any of them, and they are not orthogonal, the world is not restricted to be only in the states from an orthogonal basis. This contradicts Postulate 3, so the derivation of the Born rule from Theorem 1 does not seem to apply to SQM. The following proposition shows this.

**Proposition 1.** *If any state from a macrostate should be counted as a world, the proof of Theorem 1 cannot be used to derive the Born rule.*

The proof is given in Appendix C.

If, to keep Postulate 3, we assume that there is only a single world that is always in an ontic state, Postulate 2 will be satisfied without invoking the wavefunction collapse. However, this would be a single-world unitary theory [11–13], and this is possible only if the initial conditions are very strongly fine-tuned [14], violating Bell's statistical independence

assumption [1]. Even if this would mean something like superdeterminism, conspiracy, retrocausality, or global consistency [15], it is a possibility.

We can try a modified version of Postulate 3: "Linear combinations of ontic states can exist as long as they belong to the same macrostate. When they belong to more macrostates, collapse is invoked so that the resulting microstate is from $(|\phi\rangle)_{\phi \in \mathcal{C}}$." However, when the collapse is invoked for a measurement of $S$, and a measurement of a different subsystem $S'$ follows immediately, the subsystem $S'$ can also be in any state at the same time when the collapse is invoked for system $S$. This contradicts the modified version of Postulate 3. We can try to modify it more: "Linear combinations of ontic states can exist as long as they belong to the same macrostate. When they belong to more macrostates, collapse is invoked, but all ontic states in the macrostate that remains after the collapse are preserved." This works, but it requires the self-location interpretation of probabilities, and it would be a version of MWI where some of the worlds disappear, and the remaining ones are macroscopically indistinguishable, an ad hoc strategy. Since after recording the results of the measurements, the worlds from different macrostates no longer interfere anyway, why postulate the disappearance of some of them? It follows that the only consistent and natural way to satisfy the conditions required by the proof of Theorem 1 is the MWI. This suggests an answer to Question 7.

### 7. What Should Be Counted as a World?

The question "what should be counted as a world?" has two meanings:

Meaning 1. What kinds of unit vectors in the Hilbert space count as worlds?

Meaning 2. What components of the wavefunction should be counted when we calculate the probabilities?

However, the answer to both these questions is the same, Postulate 3.

However, since linear combinations of ontic vectors $|\phi\rangle$ from the same $\mathcal{H}_\alpha$ also belong to $\mathcal{H}_\alpha$, they are quasiclassical, and maybe they should be counted as worlds too. This happens, for example, if we try to prove the Born rule by finding a finite number of orthonormal vectors for the macrostates that add up to $|\Psi\rangle$, as in Equation (18), and counting them, as in [2,16]. If the basis $(|\phi_k\rangle)_{k \in \{1,\dots,n\}}$ from Equation (18) depends on $|\Psi\rangle$, this implies that we have to interpret all such possible orthogonal systems as consisting of words. Proposition 1 shows that this leads to overcounting, and it cannot give the Born rule. However, Theorem 1 shows that in the continuous case, if we use the same basis, in agreement with Postulate 3, this works. Therefore, Theorem 1 can be understood as the continuous limit of the proposal from [16], necessarily amended with Postulate 3.

Can Postulate 3 be avoided by defining the worlds differently?

The worlds cannot be the macrostates because this will give the naive branch counting according to which all outcomes with nonvanishing amplitude have the same probability.

Can the worlds be the nonvanishing components $\widehat{P}_\alpha |\Psi\rangle$ of $|\Psi\rangle$? It seems that they cannot be, for the same naive branch-counting argument. However, we can reinterpret probability in a decision-theoretic way as in [17,18], or as a measure of existence as in [19], or other arguments that the size of $\widehat{P}_\alpha |\Psi\rangle$ matters so that its square is the probability. It can be argued that Theorem 1 offers an alternative to these new interpretations of probability. It can also be argued that Theorem 1 is consistent with them, and it only shows that they can be understood as a coarse-graining of a more conservative probability, that of self-location in the ontic states.

### 8. The 3D Geometry as the Preferred Basis

Several important approaches to quantum gravity are background-free. We will see that background freedom brings strong evidence for the existence of an ontic basis, as in Postulate 3, but based on 3D space geometry.

Canonical quantum gravity, as formulated in [20] is based on quantizing Einstein's equation expressed in $3 + 1$ dimensions $\Sigma \times \mathbb{R}$ as in [21]. Since after quantization, time seems to disappear, the time-evolving wavefunction is decoded from the Wheeler-de Witt

constraint equation by using the Page–Wootters formalism [22]. The result is a wave-functional formulation, in which the configuration space of classical fields includes the components of the metric tensor on the 3D space $\Sigma$. The theory is invariant to diffeomorphisms, similar to gauge invariance. This makes it background-free.

The classical configuration space consists of fields $\phi = (\gamma_{ab}, \phi_1, \ldots, \phi_n) \in \mathcal{C}$, where $a, b \in \{1, 2, 3\}$, $\gamma = (\gamma_{ab})$ contains the components of the 3D metric, and $|\varphi\rangle$ represents the matter fields on $\Sigma$ and any other fields that may be needed by the theory. Let $\mathcal{C}_S$ be the configuration space of 3D metrics up to diffeomorphisms, and $\mathcal{C}_M$ the configuration space of matter fields, so that $\mathcal{C} = \mathcal{C}_S \times \mathcal{C}_M$.

A state vector with classical geometry has the form $|\Psi\rangle = |\gamma\rangle|\varphi\rangle$, where $|\varphi\rangle$ is a general quantum state of matter. Because of the invariance to diffeomorphisms, there is no correspondence between the points of $(\Sigma, \gamma_1)$ and those of $(\Sigma, \gamma_2)$, except in the special case when they are isometric. For any linear combination of states with classical geometries, there are infinitely many sets of field operators $(\widehat{\phi}_j(x), \widehat{\pi}_j(y))_j$ that satisfy the canonical (anti)commutation relations. They depend on the relative diffeomorphisms of the 3D spaces of the states in the linear combination. It is possible to fix such a set of field operators, but this would make the theory background-dependent. This is why, in background-free quantum gravity, even though the vector $c_1|\gamma_1\rangle|\varphi_1\rangle + c_2|\gamma_2\rangle|\varphi_2\rangle$ exists in $\mathcal{H}$, in general, it represents dissociated states with distinct geometries and not a superposition of two states on $\Sigma$.

This dissociation becomes even more evident if the theory of quantum gravity has a discrete 3D space or spacetime because, in this case, the underlying graphs or hypergraphs of the states in a linear combination can be nonisomorphic, so a correspondence between their points is not even possible. Examples of background-free approaches to quantum gravity in which space or spacetime is discrete include causal sets [23], Regge calculus [24], causal dynamical triangulations [25], the spin network formulation of loop quantum gravity [26,27], etc. In these approaches, the 3D space $\Sigma$ or the spacetime is a graph or a hypergraph with values attached to their vertices and (hyper-)edges to encode the metric, curvature, or spins, depending on the approach. All these approaches can be described in the Schrödinger formulation. The classical fields $\phi \in \mathcal{C}$ have to include the possible configurations of $\Sigma$. In the discrete approaches, graphs or hypergraphs representing $\Sigma$ are not assumed to be embedded in a 3D manifold. Therefore, they are background-free, in the sense that only the intrinsic properties of $\Sigma$ matter [28].

The problem of superpositions of states with different classical geometries was discussed, for example, in [29,30]. However, maybe this is not a bug but a feature of background-free quantum gravity. We claim that this dissociation leads to a new version of MWI [31].

**Observation 1.** *Due to the background freedom, linear combinations $c_1|\gamma_1\rangle|\varphi_1\rangle + c_2|\gamma_2\rangle|\varphi_2\rangle$ cannot be interpreted, in general, as superpositions.*

A state $|\Psi\rangle = |\gamma\rangle|\varphi\rangle$ with classical geometry immediately evolves into a linear combination of states with distinct geometries. This means that the basis $(|\gamma\rangle)_{\gamma \in \mathcal{C}_S}$ determines an absolute branching structure. The wavefunctional evolves on the configuration space, and its branches can interfere again. Therefore, dissociated states can reassociate. When dissociation corresponds to differences recorded at the macro level, it becomes irreversible and macroscopic branching occurs. These macroscopic differences may coincide with those due to usual branching in the MWI or may lead to additional observable effects at the macro level. This remains to be explored. As in the case of branching in Everett's interpretation, this irreversibility is not due to unitary evolution, which is reversible but to initial conditions of the universe similar to those responsible for the Second Law of Thermodynamics [4]. We will return to the problem of time asymmetry of the branching structure in Section 9.

We do not know yet if background freedom is a feature of our universe and to what extent the dissociation of the state into states with different classical geometries prevents

superposition. Probably states with different geometries that are isometric on some regions of 3D space allow for local superpositions and interference in those regions. In any case, this problem is open, and future theoretical and experimental investigations can hopefully tell us more about it. The deviations from regular quantum mechanics may be accessible to empirical testing, and experiments may corroborate or refute background freedom.

The existence of dissociation into states with different classical 3D geometries due to the absence of superpositions would make a much stronger case for the existence of ontic states. In this case, the 3D space metric of the ontic states has to be classical, so they are of the form $|\gamma\rangle|\phi\rangle$.

Whether or not quantum gravity has to be background-free in this way remains to be seen. Even if it were background-dependent, the states with classical 3D space form a special basis, consistent with our experience and with all the experiments conducted so far. Therefore, they deserve to be considered ontic states.

## 9. The 3D Geometry and the Branching Structure

To prevent violations of the Born rule in the MWI, distinct worlds should not interfere again. Branching has to occur only toward the future. It is often believed that decoherence answers Question 8, but unitary evolution is time-symmetric, so the initial conditions should break this symmetry to ensure branching only toward the future. There are strong reasons to believe that the low entropy of the initial state of the universe, postulated to explain the Second Law of Thermodynamics, also explains branching asymmetry [4]. However, we do not have a satisfactory answer for the initial low entropy either.

However, quantum gravity reveals a strong connection between the branching asymmetry and the cosmological arrow of time, i.e., the Big Bang followed by the expansion of the universe.

The Big Bang singularity consists of the fact that the 3D space metric vanishes as $t \to 0$ [32]. It is often believed that classical general relativity breaks down at singularities. However, there is a formulation of general relativity whose equations do not break down for a large class of singularities. Its equations are equivalent to Einstein's outside singularities but remain finite at singularities [33]. Such "benign" singularities require that the matter fields are constant in the directions in which the metric tensor is degenerate. This means that, since $\gamma_{ab} \to 0$ in all directions as $t \to 0$, the matter fields have to become constant on the 3D space $\Sigma$. The set of possible classical fields consistent with this condition is described by a very small number of parameters. The wavefunctional is, therefore, constrained initially to a small subspace of the Hilbert space, a single macrostate of very low entropy. The wavefunctional gradually expands and spreads over more and more, larger and larger macrostates.

This explanation makes sense even if our quantum-gravitational universe is not background-free. However, since at the Big Bang singularity, there is a unique 3D space geometry $\gamma_{ab} = 0$, the state is fully associated. Since background freedom implies that $\Psi$ dissociates as it evolves, it seems to give a stronger reason for the time asymmetry of the branching structure than the background-dependent theories.

## 10. Conclusions

We have seen that the wavefunctional formulation of quantum field theory comes implicitly with a natural interpretation of $\Psi$ in 3D space. This has implications for several different problems in quantum mechanics. The central implication is that it provides an ontology in terms of local beables. This ontology requires a preferred basis, the ontic basis. Since we can only directly observe the macrostates, the ontology of the ontic microstates justifies counting them as possible states in which the system is, just like in classical physics. However, unlike classical physics, in quantum mechanics, a state can evolve into a linear combination of microstates. The local beable ontology of the wavefunctional suggests interpreting these linear combinations as multiple ontic states coexisting in parallel. Since a macrostate is an equivalence class of microstates, probabilities arise by taking into account

the possible microstates in each macrostate. It turns out that this probability satisfies the Born rule.

If there were a single ontic world, this probability would be subjective, representing the uncertainty about the microstate. However, we have seen that, even in the standard interpretation of quantum mechanics, multiple ontic states have to coexist in parallel. Therefore, the probability should be about the self-location of the agent in one of the microstates. It follows that a new version of MWI is unavoidable in this framework. In this version of MWI, because the ontic states are orthogonal, the agent can exist only in an ontic state, and the macrostates can consist of a different amount of microstates, probabilities appear from the agent's self-location uncertainty about the microstate.

If background freedom is a feature of quantum gravity, it implies that the wavefunctional dissociates into states with distinct but classical 3D geometries. This gives strong additional support to the big picture described above. In addition, quantum gravity suggests that the Big Bang singularity may explain the time asymmetry of the branching structure because at the Big Bang singularity, the state is not dissociated, all of its components having the same geometry $\gamma_{ab} = 0$ and constant fields. As the universe evolves, it spreads over more and more macrostates, so the wavefunctional branches more and more.

**Funding:** This research received no external funding.

**Institutional Review Board Statement:** Not applicable.

**Informed Consent Statement:** Not applicable.

**Acknowledgments:** The author thanks Paul Tappenden for very helpful feedback and to the reviewers for valuable comments and suggestions offered to a previous version of the manuscript. Nevertheless, the author bears full responsibility for the article.

**Conflicts of Interest:** The author declares no conflict of interest.

## Appendix A. The Wavefunction as an Object in 3D Space

In NRQM, the wavefunction for $n$ particles is defined on the configuration space $\Sigma^n$, and it can be expressed as $n$ functions on $\Sigma$ only in the absence of entanglement.

However, in NRQM, the wavefunction is also an object of Euclidean geometry. A figure consisting of triangles and other polygons is an object of Euclidean geometry. This remains true if we label its vertices with complex numbers. $\Psi(x_1, \ldots, x_n)$ is equivalent to infinitely many figures consisting of $n$ points in $\mathbb{R}^3$, each such figure $(x_1, \ldots, x_n)$ being labeled with the complex number $\Psi(x_1, \ldots, x_n)$. We can also interpret labeled figures as unlabeled figures in a complex line bundle over 3D space [34].

The wavefunction is an object of Euclidean geometry also, according to Klein's Erlangen program [34,35]. Moreover, if we apply Klein's ideas to quantum theory and require the Hilbert space to be a representation of the Galilei group or the Poincaré group, as Wigner and Bargmann did, we get that the wavefunction is an object of spacetime, the classification of the types of particles by spin and rest mass, and the free evolution equations as in quantum theory [36–38]. For more details, see [34].

Moreover, it is also possible to represent the wavefunction as a vector field with infinitely many components on $\Sigma$. In [39], it was shown that the usual tensor product of functions defined on 3D space can be represented as a direct sum by using an additional global gauge symmetry. By direct sums between these vector bundles subject to gauge equivalence, the full tensor product Hilbert space can be represented as a vector field. Since the resulting representation is redundant, the redundancy is removed by using an even larger global gauge symmetry. Then, this global gauge symmetry can be made local by introducing a flat connection for its group. This allows the field representing $\Psi$ to be locally separable in the sense that it can be changed in an open subset $A$ of $\Sigma$ without affecting its values outside of $A$. The Hamiltonian is local, and the field evolves locally as long as no wavefunction collapse is assumed to take place.

This representation also applies to quantum field theory in the Fock representation. It is a faithful representation of $\Psi$, which can, therefore, be seen as consisting of local beables. However, this representation is artificial and was given in [39] only as a proof of concept. The natural representation is given in Sections 2 and 3.

### Appendix B. The Existence of a Measure on the Configuration Space of Classical Fields

If the configuration space of classical fields $\mathcal{C}$ were an infinite-dimensional manifold, no analog of the Lebesgue measure could be defined on it (although other measures are possible [40]). However, there are indications that the dimension of $\mathcal{C}$ is finite: the fields are constrained by equations, the gauge degrees of freedom need to be factored out, the entropy bound indicates that the Hilbert space has a finite number of dimensions in bounded regions of space [41,42], and the arrow of time requires severe additional constraints [43]. Therefore, we will assume that the manifold $\mathcal{C}$ is finite-dimensional if this is what it takes for it to be compatible with a measure $\mu$.

### Appendix C. Possible Worlds Should Form a Basis

**Proof of Proposition 1.** For every $n$, let $|\Psi\rangle = 1/\sqrt{n}\sum_{k=1}^{n}|n,k\rangle$ be a decomposition of $|\Psi\rangle$ in orthonormal vectors, so that, as $n \to \infty$, $N_{n,\alpha}/n$ converges to $\langle\Psi|\widehat{P}_{\alpha}|\Psi\rangle$, where $N_{n,\alpha} = \{k \in \{1,\ldots,n\}||n,k\rangle \in \mathcal{H}_{\alpha}\}$. Let $S_{n,\alpha}$ be the set of vectors obtained from $|n,k\rangle$ by all unitary transformations of $\mathcal{H}_{\alpha}$ that preserve $\widehat{P}_{\alpha}|\Psi\rangle$. Unitary symmetry implies that any vector from $S_{n,\alpha}$ belongs to orthogonal systems similar to $\{|n,k\rangle|k \in N_{n,\alpha}\}$. Therefore, by the hypothesis of Proposition 1, they should be counted as worlds. Let $\mathfrak{p}(\mathcal{S})$ denote the probability measure of a set $\mathcal{S} \subseteq \mathcal{H}$ of state vectors counting as worlds. Let $\alpha \neq \beta \in \mathcal{A}$ so that $|\widehat{P}_{\alpha}|\Psi\rangle| = |\widehat{P}_{\beta}|\Psi\rangle| \neq 0$. Due to unitary symmetry, there is a unitary transformation $\widehat{S}$ that maps the line $\mathbb{C}\widehat{P}_{\beta}|\Psi\rangle \subset \mathcal{H}_{\beta}$ to the line $\mathbb{C}\widehat{P}_{\alpha}|\Psi\rangle \subset \mathcal{H}_{\alpha}$, so that either $\widehat{S}\mathcal{H}_{\beta} = \mathcal{H}_{\alpha}$, or $\widehat{S}\mathcal{H}_{\beta} \subsetneq \mathcal{H}_{\alpha}$, or $\mathcal{H}_{\alpha} \subsetneq \widehat{S}\mathcal{H}_{\beta}$. The symmetry requires that $\mathfrak{p}(\widehat{S}\mathcal{H}_{\beta}) = \mathfrak{p}(\mathcal{H}_{\alpha})$. It also allows the existence of infinitely many such transformations. Let $\widehat{S}'$ be another one with the same properties so that $\widehat{S}'\mathcal{H}_{\beta} \neq \widehat{S}\mathcal{H}_{\beta}$. Since $\widehat{S}\mathcal{H}_{\beta} \cap \widehat{S}'\mathcal{H}_{\beta}$ is a strict subspace of $\widehat{S}\mathcal{H}_{\beta}$, $\mathfrak{p}(\widehat{S}\mathcal{H}_{\beta} \cap \widehat{S}'\mathcal{H}_{\beta}) = 0$, and $\mathfrak{p}(\widehat{S}'\mathcal{H}_{\beta}) = \mathfrak{p}(\widehat{S}'\mathcal{H}_{\beta} \setminus \widehat{S}\mathcal{H}_{\beta}) = \mathfrak{p}(\widehat{S}\mathcal{H}_{\beta} \setminus \widehat{S}'\mathcal{H}_{\beta}) = \mathfrak{p}(\widehat{S}\mathcal{H}_{\beta})$. Therefore, $\mathfrak{p}(\mathcal{H}_{\alpha}) > \mathfrak{p}(\widehat{S}\mathcal{H}_{\beta}) + \mathfrak{p}(\widehat{S}'\mathcal{H}_{\beta}) > \mathfrak{p}(\widehat{S}\mathcal{H}_{\beta}) = \mathfrak{p}(\mathcal{H}_{\beta})$. However, according to the Born rule, $\mathfrak{p}(\mathcal{H}_{\alpha}) = \mathfrak{p}(\mathcal{H}_{\beta})$. It follows that the Born rule is satisfied only if $\widehat{S}\mathcal{H}_{\beta} = \mathcal{H}_{\alpha}$ for every $\alpha \neq \beta \in \mathcal{A}$. However, now we will show that, for $|\widehat{P}_{\alpha}|\Psi\rangle| > |\widehat{P}_{\beta}|\Psi\rangle|$, this contradicts the Born rule. The angle $\omega_{n,\alpha}$ between $|n,k'\rangle$ and $1/\sqrt{n}\sum_{k\in N_{n,\alpha}}|n,k\rangle$ when $k' \in N_{n,\alpha}$ satisfies $\cos\omega_{n,\alpha} = |\langle n,k'|1/\sqrt{nN_{n,\alpha}}\sum_{k\in N_{n,\alpha}}|n,k\rangle| = 1/\sqrt{nN_{n,\alpha}}$. Therefore, as $n \to \infty$, $\omega_{n,\alpha} \to \pi/2$, for all $\alpha$. It follows that in the limit $n \to \infty$, $\mathfrak{p}(S_{n,\alpha})/\mathfrak{p}(S_{n,\beta}) = 1$. Therefore, counting all vectors from the sets $S_{n,\alpha}$ as worlds contradicts the Born rule. $\square$

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
