# Peer review of "The Relation between Wavefunction and 3D Space Implies Many Worlds with Local Beables and Probabilities"

_quantumrep, doi:10.3390/quantum5010008_

Round 1
Reviewer 1 Report
This is an interesting and original paper. Some of the material in the paper is beyond my expertise. I suggest sending the paper to additional experts. However, I have a number of questions for the author(s) the answers to which may improve or at least clarify the paper.
Page 2: I'm not sure how to understand the notion of macrostates used in Postulate 2 and its consistency in quantum mechanics with Postulate 3: Suppose that a system is found by a measurement to be a macrostate defined by the corresponding macro-projection operator (say after a measurement of the corresponding macro-observable). What is the quantum state of the system? Is it one of the ontic states, or is it a superposition of all ontic states belonging to this subspace?
The author(s) seem (s) to say that the ontic states are classical. By this I understand that they mean that the state of the system is one of the states belonging to the subspace. It seems to me that such an interpretation is consistent with quantum mechanics only for certain cases in which prior to the measurement the system was prepared in one of these degenerate eigenstates of the macro-projector. I don't see or at least did not understand the assumptions added over and above standard quantum mechanics that lead to the author(s) conclusion.
Page 9: The authors seem to argue that background freedom supports the result that the 3D geometry provides the preferred basis for pure quantum mechanics in the sense that only 3D geometries are ontologically distinguished. This should be explained a bit more slowly: It seems to me that this implies that at some level of description quantum mechanics is violated. Is this true?
Page 10: The author(s) argue that superposition of states with different classical geometries lead to absolute decoherence. I am not sure exactly what absolute decoherence means. If the idea is the standard one according to which decoherence means the decay of interference terms, do the authors mean here that the interference terms become zero at finite times? How is this possible if the time evolution is unitary?
Page 10, a bit below: The authors say that when dissociation becomes irreversible, macroscopic branching occurs. I did not understand, what "irreversible" means here, given that the time evolution is unitary.
Author Response
I wish to thank the reviewers for valuable feedback which improved the clarity and quality of the manuscript.
> Page 2: I'm not sure how to understand the notion of macrostates used in Postulate 2 and its consistency in quantum mechanics with Postulate 3: Suppose that a system is found by a measurement to be a macrostate defined by the corresponding macro-projection operator (say after a measurement of the corresponding macro-observable). What is the quantum state of the system? Is it one of the ontic states, or is it a superposition of all ontic states belonging to this subspace?
> The author(s) seem (s) to say that the ontic states are classical. By this I understand that they mean that the state of the system is one of the states belonging to the subspace. It seems to me that such an interpretation is consistent with quantum mechanics only for certain cases in which prior to the measurement the system was prepared in one of these degenerate eigenstates of the macro-projector. I don't see or at least did not understand the assumptions added over and above standard quantum mechanics that lead to the author(s) conclusion.
It is a linear combination of ontic states belonging to this subspace. Not just an ontic state, this would not work, as the reviewer pointed out.
While I returned later in the paper to explain this, I agree with the reviewer that it is not stated in Postulate 2. In fact the reviewer's feedback made me realize that I had to complete Postulate 2, so I extended it with an additional part (ii) (lines 173-174).
If we assume that projection happens, the result is the projected state vector (normalized). If we allow unitarity as in MWI, the subsequent observations are consistent with the state vector being projected, even if it is not.
My initial formulation of Postulate 2 was intended to avoid making claims beyond what we can observe (e.g. whether there is a collapse or just unitary evolution). But even with part (ii) Postulate 2 makes fewer assumptions than the Projection Postulate. I added an explanation of why Postulate 2 avoids assuming too much in lines 175-179.
What I mean is that the projection of Psi associated to the macrostate consists of ontic (classical) states. Mathematically it can be any linear combination of ontic states from that macrostate, this is why I say they are the microstates. But in Section 3 I interpret this as a set of classical states, aggregated by a measure.
I also added more explanations about quantum measurements in lines 191-196, after the more detailed description that was there in lines 180-190.
I added an explanation of the consistency between Postulates 2 and 3 in lines 203-207.
> Page 9: The authors seem to argue that background freedom supports the result that the 3D geometry provides the preferred basis for pure quantum mechanics in the sense that only 3D geometries are ontologically distinguished. This should be explained a bit more slowly: It seems to me that this implies that at some level of description quantum mechanics is violated. Is this true?
Indeed, it may lead to different predictions compared to regular quantum mechanics, but I cannot say at this time how strong these differences are. In lines 423-430 I added an explanation that these are open problems.
> Page 10: The author(s) argue that superposition of states with different classical geometries lead to absolute decoherence. I am not sure exactly what absolute decoherence means. If the idea is the standard one according to which decoherence means the decay of interference terms, do the authors mean here that the interference terms become zero at finite times? How is this possible if the time evolution is unitary?
Another good observation by the reviewer. My formulation was too vague, what I wanted to say is that dissociation is an alternative to decoherence that is similar to decoherence. By "absolute" I had in mind to contrast this with those approaches where it is said that decoherence selects a preferred basis, which I therefore consider to be "relative" (because different initial states may make emerge different bases arXiv:2102.08620). But I think I should not refer to dissociation as being a kind of decoherence, because they are different, so I removed the two places where I use the words "an absolute form of decoherence".
> Page 10, a bit below: The authors say that when dissociation becomes irreversible, macroscopic branching occurs. I did not understand, what "irreversible" means here, given that the time evolution is unitary.
I am not using irreversible to refer to the dynamics, just to the fact that the dissociated states from two different macrostates no longer associate. It is just irreversibility of branching in MWI, and similar to thermodynamics irreversibility even if the dynamics is deterministic. I added explanations in lines 415-422.
Thank you for the feedback, it was very helpful!
Reviewer 2 Report
Review on article quantumrep-2197161
This paper deals with an important question: the justification of the foundations of quantum mechanics and, in particular, its many worlds interpretation. It uses well-known and proven mathematical apparatus and is correctly and clearly written.
However, the paper has one significant defect:
The abstract is poorly written -- it is very long and because of this it becomes not very clear. The abstract for a paper should be compact and clear at the same time. It should briefly list the results obtained. Everything else: explanations, justifications and so on should be contained in the introduction and other sections of the paper.
This paper can be published in the journal "Quantum Reports" once this defect is eliminated.
Author Response
I wish to thank the reviewers for valuable feedback which improved the clarity and quality of the manuscript.
> The abstract is poorly written -- it is very long and because of this it becomes not very clear. The abstract for a paper should be compact and clear at the same time. It should briefly list the results obtained. Everything else: explanations, justifications and so on should be contained in the introduction and other sections of the paper.
> This paper can be published in the journal "Quantum Reports" once this defect is eliminated.
I completely rewritten the abstract. Now it is shorter and I hope it is clear. It mainly lists the results presented in the manuscript, but for a couple of them I also briefly specified how I got them, since the method is different from other proposals from the literature.
Round 2
Reviewer 1 Report
I am fine with the author's responses and revisions to my previous comments.
One last point: I'm not sure what the author means by the remark on Page ??? that Postulate 2 avoids the assumption that the wavefunction collapses -- -- without forbidding it. I am not sure what is meant by this last clause.
Typo in Line 436 which says: But even if were background-dependent
Author Response
I fixed the typo on line 436.
On line 217 I replaced "it avoids the assumption that the wavefunction collapses (without forbidding it)" with "it avoids presuming whether the wavefunction collapses or not". That is, if we denote the proposition P="the wavefunction collapses", Postulate 2 replaces the Projection Postulate in a way that doesn't assume P and it also doesn't assume not P.